# The Effect of Therapeutic Hypothermia after Cardiac Arrest on the Neurological Outcome and Survival—A Systematic Review of RCTs Published between 2016 and 2020

**DOI:** 10.3390/ijerph182211817

**Published:** 2021-11-11

**Authors:** Christian Colls Garrido, Blanca Riquelme Gallego, Juan Carlos Sánchez García, Jonathan Cortés Martín, María Montiel Troya, Raquel Rodríguez Blanque

**Affiliations:** 1Intensive Care Unit, Hospital Universitario Clínico San Cecilio, 18071 Granada, Spain; christian.16.ef@gmail.com; 2School of Nursing, Faculty of Health Sciences, University of Granada, 18071 Granada, Spain; jsangar@ugr.es (J.C.S.G.); jonathan.cortes.martin@gmail.com (J.C.M.); 3Instituto de Investigación Biosanitaria, ibs.GRANADA, 18012 Granada, Spain; 4Research Group CTS1068, Andalusia Research Plan, Junta de Andalucía, 18014 Granada, Spain; mariamontiel@ugr.es (M.M.T.); raquel.rodriguez.blanque.sspa@juntadeandalucia.es (R.R.B.); 5School of Nursing Ceuta Campus, Faculty of Health Sciences, University of Granada, 51001 Ceuta, Spain; 6Distrito Sanitario Granada-Metropolitano, 18013 Granada, Spain

**Keywords:** induced hypothermia, cardiac arrest, survival rate, critical care

## Abstract

Therapeutic hypothermia is a treatment used for patients who have suffered cardiorespiratory arrest and remain conscious after the recovery of spontaneous circulation. However, its effectiveness is controversial. The objective of this systematic review is to summarize the scientific evidence available about the effect of therapeutic hypothermia on neurological status and survival in this type of patients. Methodology: A primary search in CINAHL, CUIDEN, Pubmed, Web of Science, and Scopus databases was carried out. Randomized clinical trials (RCT) published from 2016 to 2020 were selected. Results: 17 studies were selected for inclusion and most relevant data were extracted. Methodological quality was assessed by the RoB tool. Conclusions: Although therapeutic hypothermia is a safe technique with few adverse and manageable effects, it has not shown to improve survival rate and neurological status of adult nor pediatric patients. It is possible that its positive effect on neuroprotection could be achieved only by preventing hyperthermia although further investigation is needed.

## 1. Introduction

Cardiorespiratory arrest (CA) is one of the main causes of death in developed and developing countries. CA is defined as the cardiac failure caused by the cessation of mechanic activity, which is confirmed by poor circulation. The cardiopulmonary resuscitation (CPR) procedure aims to restore circulation by ensuring ventilation and gas exchange through CPR and defibrillation, among other emergency techniques [1].

Worldwide out-of-hospital cardiorespiratory arrest (OHCA) rate is 83.7 cases per 100,000 per year. In adults, the incidence increases to 95.9 cases per 100,000 annually. These numbers vary between countries and are higher in North America (98.1) or Australia (112.9) in comparison with Asia (52.5). Annually, 300,000 OHCA take place in North America and 275,000 approximately in Europe. Most of them occur at home (66%), followed by workplaces and public spaces (20%). The average age of patients that suffer a CA is 71 years, being more frequent among men than women [2,3].

The main causes of CA are cardiopulmonary (43%), chronic pulmonary affection (13%), diabetes (13%), and expected death (5%). In 80.3% of cases, asystole is present, while signs of ventricular tachycardia or ventricular fibrillation and electromechanical dissociation are less common (5.9% and 6%, respectively) [4]. Only 22.8% of all patients who suffer a CA manage to recover spontaneous circulation. The survival rate of these patients is 4.9% at 30 days, which increases to 10.4% when resuscitation is initiated immediately. If CA takes place in a hospital environment, these rates increase to 22%, and mortality prevalence for patients that have recovered spontaneous circulation after a CA and are admitted to hospital is 57% [5].

In this context, post-resuscitation care after spontaneous circulation recovery becomes vital due to the associated high intra-hospital mortality [6].

Post-resuscitation care aims to control the problem behind the CA, reducing the damage caused by post-CA syndrome. It focuses on neuroprotection in order to prevent further neurological damage after the arrest [7]. More than 50% of these patients show a sub-optimal neurological status at discharge [8,9]. One of the interventions that appears to be effective to prevent further brain damage is targeted temperature management (TTM), previously known as therapeutic hypothermia (TH) [10,11]. It consists on the application of cold through different devices that reduce the patient’s temperature in a controlled and gradual manner [12]. It decreases brain metabolism by reducing oxygen and glucose consumption by brain cells [13]. Thus, it modulates the inflammatory response induced during the reperfusion stage by stabilizing enzymatic reactions. In this phase, reactive oxygen species and neurotransmitters are released, generating excitotoxicity [14], so TTM decreases intracranial pressure by enzyme stabilization and vasoconstriction [12]. It is currently also used in cases of anoxic brain injuries, mild brain injuries, stroke, hypoxic ischemic encephalopathy, bacterial meningitis, post-surgical tachycardia, and acute respiratory distress syndrome [15]. When a CA is ongoing, the therapeutic decision to apply TTM must be taken quickly and promptly since it has been shown that its beneficial effects are time dependent. On the other hand, TTM can be applied safely in combination with other interventions for CA, such as percutaneous revascularization [16]. The main objective of this systematic review is to summarize the scientific evidence about TTM effectiveness on neurological status and survival rate in patients who have suffered CA.

## 2. Materials and Methods

A review of TTM studies was carried out to address our scientific question according to Preferred Reporting Items for Systematic reviews and Meta-Analyses (PRISMA). The protocol is registered in PROSPERO coded as CRD42020207405 and available at https://www.crd.york.ac.uk/prospero/display_record.php?ID=CRD42020207405 (accessed on 6 November 2021).

Randomized clinical trials (RCTs) relating to TTM and temperature control interventions in patients who suffer a CA and its influence on survival and neurological status were included.

CINAHL, CUIDEN, Scopus, Cochrane, Pubmed, and Web of Science databases and Dialnet journal were consulted. The search was limited to articles written in English or Spanish, and only articles published in the last 5 years were included.

The following MeSH (Medical subject Headings) terms were used to conduct the search strategy: “hypothermia, induced”, “induced hypothermia”, “heart Arrest”, “cardiac arrest”, “survival”, “survival rate”, “survival analysis”, “mortality”. In addition, some synonyms were used: “therapeutic hypothermia”, “induced mild hypothermia”, “targeted temperature management”, “cardiac arrest”, “cardiopulmonary arrest”, “cardiorespiratory arrest”, and “neurolo*”. The Boolean operators “AND” and “OR” were used. Therefore, the complete search strategy with the Boolean operators was ((“therapeutic hypothermia” OR “hypothermia, induced” OR “induced mild hypothermia” OR “targeted temperature management”) AND (“cardiac arrest” OR “heart arrest” OR “cardiopulmonary arrest”) AND (“survival” OR “Survival rate” OR “mortality” OR “survival analysis” OR “neurolo *”)) OR ((“therapeutic hypothermia” OR “induced hypothermia” OR “mild induced hypothermia”) AND (“cardiac arrest” OR “cardiorespiratory arrest”) AND (“survival” OR “survival rate” OR “mortality” OR “survival analysis” OR “neurolo *”)).

The inclusion criteria were: RCTs carried out in patients treated with TTM after CA published in the last 5 years (1 January 2016 to 18 April 2020), written in English or Spanish. Animal studies, ongoing RCTs, unavailable full text, TTM studies that did not focus on survival rate and neurological status in patients that suffered CA, systematic reviews, meta-analysis, protocols, and observational and descriptive studies were excluded.

Our electronic search identified 606 potentially relevant studies. Once duplicates were removed (N = 260), 211 studies were selected as potentially relevant and reviewed by title and abstract. Seventeen studies were finally included according to the inclusion criteria for full reading and data extraction [17] (Figure 1).

In order to assess the methodological quality of the selected studies, we assessed the risk of bias in all included studies in duplicate by two independent reviewers (B.R.G. and J.C.S.G.) using the Cochrane risk of bias assessment tool. The following items were assessed: randomization and sequence generation, allocation concealment, blinding and performance, outcome assessment, completeness of outcome data, and selective outcome reporting. Unblinded studies were not penalized in the risk of bias assessment due to the nature of the treatments that makes blinding non-feasible [18].

The information was collected and divided into two groups: adult and pediatric patients. Studies were examined in depth, and the most relevant data were extracted: title, authorship, year of publication, sample size, main objective, treatment in intervention, and control groups and outcomes.

## 3. Results

In total, 17 randomized trials reporting on 5813 adults and 712 children were included; Table 1 and Table 2 show the characteristics of the RCTs carried out in adult and pediatric patients, respectively. Table 3 and Table 4 summarize the main results in adult and pediatric patients, respectively. The temperature used in the most frequent experimental group was 33 °C except in the study of Pang et al. (2016), Scales et al. (2017), and Look et al. (2018), which applied an experimental temperature of 32° and 34°, respectively. The neurological outcome was assessed though the Cerebral Performance Category scale (CPC) in the majority of trials conducted in adult population and though Vineland Adaptive Behavior Scales (VABS-II) in the pediatric population. The most commonly applied intervention was the combination of invasive and external cooling in adult patients with the exception of Look et al.’s (2018) and Nordberg et al.’s (2019) studies, which only used external cooling. The time after which the survival rate was analyzed was very heterogeneous. It was determined after hospital discharge in three studies Bernard et al. [18], Look et al. [19] and Scales et al. [20] after one month of follow-up in Meert et al. [21], after three months of follow-up in Lascarrou et al. [22], Lopez-de-Sa et al. [23] and Nordberg et al. [24], after six months of follow-up in Cronberg et al. [25], Kirkegaard et al. [26], Lilja et al. [27], Pang et al. [28] and Fink et al. [29], and after one year after suffering CA in Maynard et al. [30], Moler et al. [31] Moler et al. [32], Silverstein et al. [33]. The quality of included studies was overall moderate, with some studies demonstrating a high risk of bias (Figure 2). One study had a high risk of bias for randomization, and 10 (10/17, 58.82%) had some concerns. Two studies had a high risk of bias for randomization, and six had some concerns. One study had a high risk of bias for allocation concealment. Outcomes assessment (i.e., attrition) was judged to have a high risk of bias in two studies and was inadequate in three studies but good in 12 studies (12/17 70.58%). Thirteen studies had a low risk of bias for detection (i.e., selective reporting), and 10 had some concerns in outcomes reporting (i.e., incomplete data).

## 4. Discussion

### 4.1. Therapeutic Hypothermia in Adults

Cardiorespiratory arrest data in adults was homogeneous. CA patients were aged between 61 and 68 years. Most studies reported a higher proportion of men (74.8–85.7%) [19,24,25,26,27] Nonetheless, in other studies, that proportion dropped to 65% [20,22]. Two studies highlighted that 60% of cases of cardiorespiratory arrest are witnessed by someone [19,20]. In contrast, in two studies, this percentage rose to 90% [22,27]. However, only 46% of CA are attended by witnesses or non-medical personnel before emergency services arrival [20]. Regarding the etiology of CA, 85–96% present a probable cardiac cause [18,19]. However, the study by Lascarrout et al. that analyzed non-shockable rhythms CA reported asphyxia as the main cause of CA in 56% of cases [22].

A total of 41% to 47% of patients showed shockable rhythm when a defibrillator was connected [19,20,24]. Cronberg et al. reported that approximately 93–94% of patients who suffered a CA and survived at least six months had showed an initial shockable rhythm [25]. In addition, Lascarrout’s study showed that only 17.7% of patients who suffered a CA with an initial non-shockable rhythm were alive at 90 days [22]. Furthermore, non-shockable rhythms are associated with worse neurologic status [36]. Results from another study stated that those patients that were defibrillated by a witness of the CA increased by 67% the chance of ending with a good neurological state at discharge [23]. In this context, early defibrillation is essential since it is the most effective treatment against ventricular fibrillation and increases survival rates of CA patients [37]. A rang of 54.4–57% of the reported CA cases occurred at home and 38.7% in a public place [26,27]. Studies indicated that angiography procedure appears to be the most frequent post-CA intervention (56–82% of cases). TTM can be applied simultaneously with angiography, so the application of one technique should not be delayed by the other [26,27].

### 4.2. Targeted Temperature Management

Results from the study by Lascarrout et al. reported better neurologic status 90 days after resuscitation in CA patients treated with TH at 33 °C compared to the control group using the CPC Scale [22]. In accordance with this evidence, patients with initial non-shockable rhythms may also benefit from TH. However, in this study, patients in the control group were not excluded if they experienced fever (temperature >38 °C). This fact could be responsible for harmful effects, thereby overestimating the beneficial effects of TTM. Hyperthermia prevention could have been enough to achieve these benefits.

In contrast, Lilja et al. compared the effectiveness of TH at 33 °C and 36 °C, concluding that no benefits on neurologic status were found according to the CPC scale [27]. Moreover, these results were in accordance with another study that compared TH at 32 °C, 33 °C, and 34 °C, and no differences in neurologic status were found [27]. It is worth mentioning that the sample size in that study was a great deal smaller, whereby results should be analyzed with caution.

Cronberg et al. concluded that there were no significant differences in any of the scales that measured cognitive function, quality of life, and neurological status in patients who had suffered a CA and had been treated with TH at 33 °C or 36 °C [25].

Therefore, the evidence appears to indicate that the prevention of hyperthermia and TTM at 36 °C are more effective than TH application. Nonetheless, it is possible that specific subgroups of patients that have suffered a CA may benefit from TH.

#### 4.2.1. Adverse Events

As the Lopez-de-Sa et al. study indicated, the group assigned to TH at 32 °C showed a lower risk of respiratory infections, however, they suffered arrhythmias more frequently, specifically bradycardias [35]. Results from another RCT came to similar conclusions when patients treated with TH at 32 and 34 °C were compared. Patients assigned to the 32 °C TH group showed lower prevalence of seizures but more cases of bradycardia [38]. Nevertheless, one study concluded that bradycardia during TH intervention was associated with better outcomes at hospital discharge [39].

On the other hand, another frequent adverse effect associated with TH reported by Mérei et al. is the likelihood of developing thrombocytosis [35].

These adverse events are in line with the current evidence [16]. Additionally, it has been shown that these events have no serious health consequences, so TH is considered a safe therapy, and it can be implemented together with other interventions, such as extracorporeal life support (ECLS), as Pang et al. reported [28].

Bernard et al. concluded that patients that received TH during their CA showed higher rates of pulmonary edema (10%) than those who received standard treatment (4.5%) [19]. Thus, ice-cold saline for the induction of mild hypothermia is discouraged during CA. Another study pointed that saline infusion leads to a rise central venous pressure (CVP), compromising coronary perfusion [40].

On the contrary, Scales et al. reported no adverse effects associated to TH application five minutes after CA. Furthermore, pulmonary edema was lower in the group that received out-of-hospital TH [20]. One explanation could be that only 72% received intravenous out-of-hospital TH and that the volume infused was smaller than in the Bernard et al. study.

None of the studies included in this review showed significant differences in survival rate.

#### 4.2.2. Prognostic Factors

Mérei et al. observed higher rates of mortality in patients with lower Glasgow Coma scale (GCS) score at the time of admission to intensive care [35]. They concluded that neurologic status at discharge was a long-term indicator of neurologic status and survival [37]. This could mean that the chances of improving neurological status of CA patients is very low.

#### 4.2.3. Quality of Life in CA Patients

Cronberg et al. reported in their study that 49.4% of CA survivors were active workers before the event, and only 30% continued working after the event [25]. Therefore, it can be concluded that surviving a CA will likely affect the daily life of these patients due to CA sequelae.

### 4.3. Targeted Hypothermia in Out-of-Hospital CA Patients

The devices most frequently used in the studies analyzing TH effectiveness in the out-of-hospital environment are cold saline infusion and intranasal evaporative catheter.

Bernard et al. reported that patients who received cold saline infusion during CA took more time to return to spontaneous circulation than those who underwent standardized treatment. In addition, mortality rates were higher in CA patients treated with TH during CA compared to those who received normal treatment. However, no significant survival differences were found between groups [19]. One possibility is that saline infusion TH treatment could be delaying other procedures aimed at CA reversion. These findings are in line with the study by Maynard et al., where they concluded that patients that received out-of-hospital TH showed similar outcomes in terms of neurological status to patients that were treated with TH after hospital arrival [30]. Moreover, patients who received out-of-hospital TH were more likely to also receive hospital TH according to Scales et al. Nonetheless, this fact did not lead to higher survival rates or better neurologic status [20]. In contrast, the study carried out by Garrett et al. showed that 36.5% of the CA patients that received TH recovered spontaneous circulation compared to 26.9% of those who did not receive it. This, however, did not lead to a rise in survival rate [41].

Nordberg et al. concluded that using an intranasal device during CA could reduce body temperature to the target temperature faster than standard care. However, survival rate and neurological outcomes were similar between groups in this case, too [24].

Considering the results of these studies, it seems more reasonable to initiate TH using intranasal devices rather than cold saline infusion. Moreover, these findings also indicate that TH may possibly not have the beneficial effect on patients that Holzer et al. and Bernard et al. reported in their studies back in 2002 [42,43].

#### 4.3.1. Maintenance Phase

Kirkegaard et al. reported no differences in survival rates between patients treated with TH during 24 or 48 h. However, patients that showed optimal neurologic status at six months after suffering a CA was 5% higher in the 24-h TH group. Although no significant differences were found, it is not possible to completely dismiss possible beneficial effects. In addition, that study showed higher prevalence of adverse events such as hypotension in the 48-h TH group [26]. Another observational study concluded that increasing TH time led to higher rates of hemorrhage, pneumonia, and arrhythmia [44]. Therefore, a 24-h TH intervention seems more effective.

#### 4.3.2. Internal and External Cooling Devices

The study by Look et al. showed that patients treated with internal devices reached higher survival rates than patients that did not receive TH. They also found that those devices allowed for a more accurate temperature control. Nonetheless, survival rates were not statistically different and the insufficient sample size ought to be taken into consideration [34].

Another observational study came to a similar conclusions, reporting no significant differences in survival rates in terms of neurological status between patients that received invasive TH in comparison to surface cooling with blankets or ice [45]. Another study also pointed out that invasive device use increases the risk of infection and bleeding [46]. Therefore, internal device use entails a higher risk of adverse events despite offering a more accurate temperature control.

#### 4.3.3. Risk of Bias

In order to carry RCTs on TTM, a large enough sample size is required since TH beneficial effect rates may not be higher that 5–10%. In this context, this kind of RCTs are usually multicenter, but in some cases, the choice of certain TH parameters are left to the individual centers, which can lead to important differences between hospitals. In the cases of Kirkegaard et al. and Lascarrout et al., different cooling devices were used in the TH intervention, which could lead to biased results [22,26]. Moreover, studies like those by Mérei et al. or Scales et al. have an insufficient sample size [20,35]. On the other hand, it is not possible to interview all CA survival patients to assess neurologic status at discharge. In the study by Maynard et al., for example, the proportion of patients interviewed was only 73% [37]. Bernart et al. had to end the study when targeted temperature protocol at the hospitals participating in the trial changed from 33 °C to 36 °C. That change was imposed after TTM study publication [47]. Subsequently, lower rates of compliance of TH at 33–36 °C were reported, as were higher rates of fever and worse clinical prognosis. We assessed the risk of bias using the Cochrane risk of bias assessment tool (Higgins et al., 2011), which demonstrated moderate risk of bias in the majority of included studies [18].

As a result, Bray et al. recommended that hospitals that decide to practice TH at 36 °C should adequately sedate patients and consider the use of muscle relaxants in order to prevent fever onset [48].

The impossibility of blinding the TH interventions is another determinant of bias.

Finally, the most commonly used scales to measure neurologic status in this kind of patients are CPC and mRS. Rittenberger et al. concluded that those instruments have not been validated, and more precise tools are needed to assess CA survivors [49].

### 4.4. Hypothermia in Pediatric Patients

Mean age of pediatric patients that suffered CA was between 1 and 3.2 years old [30,31,32,33]. Most of the studies showed a higher proportion of males (59–66%) [21,31,32]. However, the study by Fink et al. included 41% of males [29]. The main cause of CA was respiratory (72–88%), with this etiology being more frequent in the pediatric population than in adults. A cardiac etiology was observed in 10% of cases [29,32]. From the studied CA events, 11–38% were witnessed, and in 65% of cases, resuscitation was initiated by non-medical professionals [21,32]. Non-shockable initial rhythms were the most frequent (72–88% of cases). Shockable rhythms constituted between 3–7%, and in 6–15%, bradycardia was detected [29,31,32]. The study by Moler et al. (2017), which analyzed intra-hospital CA, found that 57% of patients presented bradycardia, 21% pulseless electrical activity (PEA), 8% asystole, and 10% ventricular fibrillation (V-fib) [32]. One hypothesis is that intra-hospital CA are treated immediately, whereas in out-of-hospital Cas, the connection to a defibrillator is somewhat delayed. It should be noted that, according to the results of Moler et al., the main cause of mortality after recovery of spontaneous circulation in out-of-hospital CA is brain death or withdrawal of life support due to poor neurological prognosis in 81% of cases. That percentage, however, drops to 36% in the intra-hospital environment. Furthermore, survival rate with good neurological status was 16% in out-of-hospital CA patients and 38% hospital patients. This demonstrates how the prognosis of a CA is intimately related to the time between onset of the arrest and initiation of resuscitation measures [31,32].

According to the study carried out by Pittl et al., factors associated with survival rate in pediatric patients were cardiac etiology, ventricular fibrillation initial rhythm, shorter resuscitation, lower epinephrine dose, and weekday and daytime events [46].

#### 4.4.1. Targeted Temperature Management (TTM)

In the studies by Moler et al. comparing out-of-hospital and intra-hospital TTM at 33 and 36.8 °C in CA patients, no differences were found in one-year survival rates nor neurological status [31,32]. Meert et al. came to the same conclusion when TTM at 33 °C for 48 h was applied to the intervention group in comparison to the control group, which received normothermia [21].

#### 4.4.2. Adverse Events

Meert et al. reported a lower proportion of patients that required transfusion of blood products among those treated with TTM although they suffered a higher prevalence of hypoglycemias [21]. Nonetheless, these results are probably unreliable, as the sample size was not statistically valid. Furthermore, other studies with an adequate sample size showed no differences in terms of adverse events or greater use of blood products [31,32].

#### 4.4.3. Maintenance Phase Duration

The results of the study by Fink et al., which compared TTM for 24 and 72 h, showed a decrease of brain damage biomarkers in the 72-h treatment group. Nevertheless, no differences in survival rates and levels of brain damage were found after six months of follow-up [29]. These findings, notwithstanding, should be taken with care since there were already differences in these biomarkers before the TTM for 24 h intervention, so its variability cannot be attributed to the treatment. In addition, this study has sampling limitations, too, so its conclusions cannot be extrapolated.

#### 4.4.4. Risk of Bias

It has been observed that studies in pediatric populations have much smaller sample sizes than studies in adults, which makes it complicated to generalize their conclusions and increases the risk of bias. This fact also limits the ability to detect the beneficial effects of TTM if rates are low (Figure 2)

This review has certain limitations. On the one hand, the lack of RCTs with a sufficient sample size makes it difficult to reach reliable conclusions on the efficacy of TRM. In addition, it is a systematic review of RCTs published between 2016 and 2020, which may limit the results presented. Nevertheless, it has been observed through the results from previous years in the different databases that the largest number of publications occurred in the inclusion period, except for 2015.

## 5. Conclusions

The evidence on TTM suggest that it is a safe procedure with few and manageable adverse effects in the hospital environment. Furthermore, it can be used in conjunction with post-resuscitation procedures, such as angiography and extracorporeal life support. Regarding TTM use in adult population after CA, no differences were found in comparison with normothermia. However, it is possible that TTM could have beneficial effects on certain subgroups of patients. Studies in pediatric patients did not report significant differences between TTM and normothermia effectiveness even if applied for more than 24 h. However, some studies point that TTM could have a beneficial effect on survival rates and neurological status in both adults and children. Evidence shows that neuroprotection can likely be achieved solely by hyperthermia prevention although more RCTs and meta-analysis of RCT with greater sample sizes are needed to determine if differences exist. Regarding out-of- hospital TTM, the results are controversial since pulmonary edema is associated with intravenous cold saline infusion for hypothermia induction. Intranasal device use appears to be a safer method. Moreover, any type of out-of-hospital TTM improves the chances of hospital TTM intervention at admission. However, its effect on survival and neurological status has not been demonstrated. Finally, more accurate scales to assess neurologic status are needed in order to reduce variability and provide more precise results.

## Figures and Tables

**Figure 1 ijerph-18-11817-f001:**
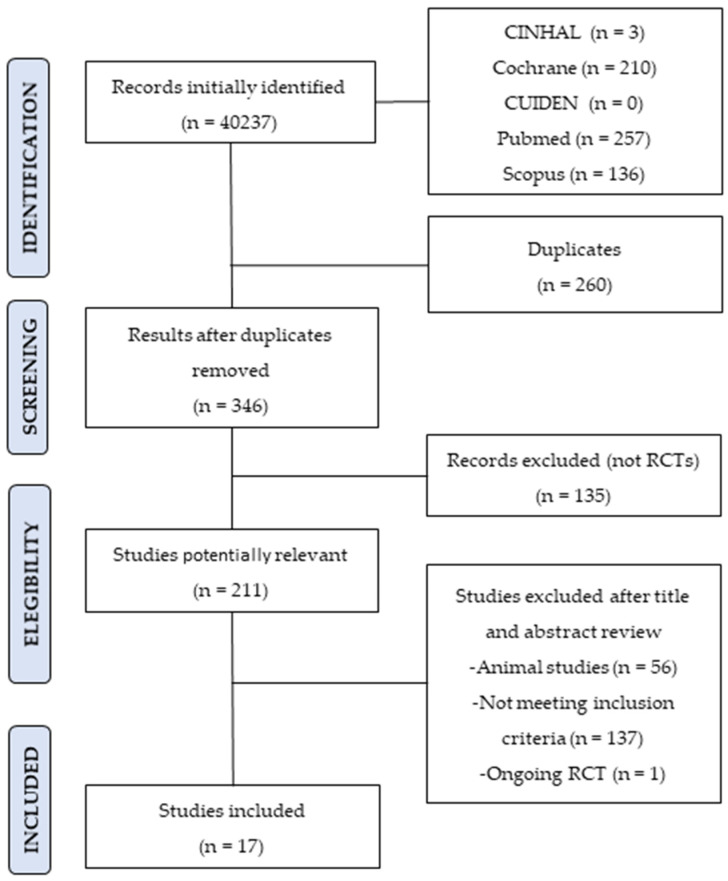
Selection flow diagram.

**Figure 2 ijerph-18-11817-f002:**
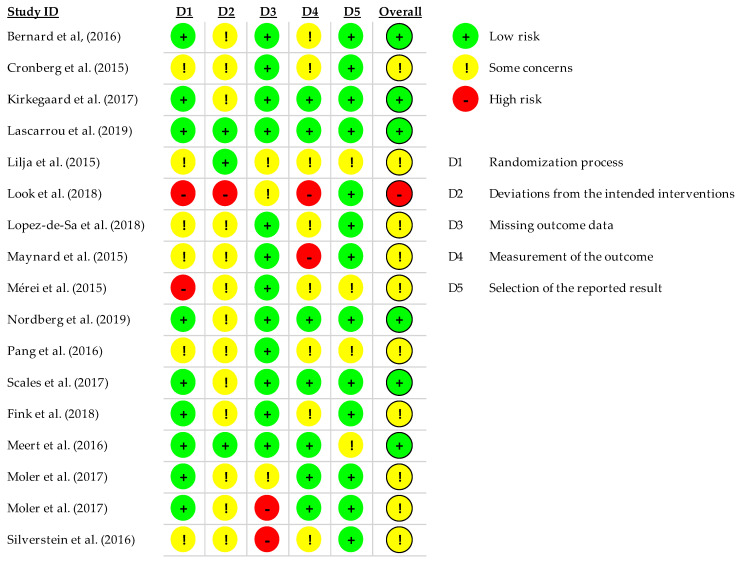
Risk of bias in included trials on effectiveness of therapeutic hypothermia after cardiac arrest on the neurological outcome.

**Table 1 ijerph-18-11817-t001:** Characteristic of the selected RCTs carried on in adult patients.

Author (Year)	N	Baseline Characteristics	Cooling System	Mean Time Reaching Target Temperature	Target Temperature	Inclusion Criteria	Exclusion Criteria	Time of Evaluating Survival	Time of Evaluating Neurological Outcome
Bernard et al. (2016) [19]	1198	≥18 years, IV access, and were still in CA after CPR	Rapid IV infusion of large-volume (30 mL/kg), cold saline vs. Standard Care	22.6 ± 11.5 vs. 20.0 ± 10.6 min	33 °C in the ICU	≥18 years of age, in cardiac arrest, had IV access established, and still in CA result of trauma (including hanging), suspected of intracranial bleeding	Pregnant women, to be already hypothermic (<34.5 °C), or inpatients in a hospital and documenting limitations in resuscitation.	Hospital discharge	Serum neuron-specific enolase concentration at 24 h
Cronberg et al. (2015) [25]	939	Unconscious (GCS < 8)	Ice-cold fluids, ice-packs, and intravascular or surface T° management devices at the discretion of the site.	20 (15–27) vs. 20 (14–30) min	33 °C	≥18 years of age, OHCA of cardiac cause with ROSC	ROSC > 240 min, CA with asystole as the initial rhythm, acute intracranial hemorrhage or stroke, and T° < 30°C	6 months	6 months
Kirkegaard et al. (2017) [26]	355	OHCA of cardiac origin	Invasive cooling with intravascular catheter	48-h group: 281 (IQR, 217–360) vs. 24-h group: 320 (IQR 241–410) min (*p* = 0.01)	33 °C	17–80 years, ROSC > 20 min, GCS > 8, shockable and non shockable rhythms	Patients with unwitnessed asystole	6 months	6-months (CPC)
Lascarrou et al. (2019) [22]	584	Iintrahospitalary: 27.4%; OHCA: 72.6%; non-cardiac cause: 66%; circulatory shock: 58%	NR	NR	33 °C ± 0.5 °C for 24 h	>18 years, resuscitated from OHCA with a non shockable rhythm, GCS > 8	Low-flow > 60 min; hemodynamic instability, CA > 300 min, moribund condition, cirrhosis, pregnancy, previous inclusion in RCT, no health insurance	3 months	3 months (CPC)
Lilja et al. (2015) [27]	652	Unconscious, ≥18 years, OHCA of cardiac origin, Controls were patients with STEMI with percutaneous coronary intervention	4 °C intravenous solutions, ice-packs, and cooling devices	20 (15–30) 21 (13–30) min	33 °C for 24 h	>18 years, OHCA of cardiac cause, unconsciousness after ROSC	Bleeding diathesis, acute intracranial bleeding, stroke, asystole, therapy limitations and Do Not Resuscitate order, CPC = 3 or 4, T° < 30 °C, SBP < 80 mmHg	3 months	3 months
Look et al. (2018) [34]	87	Controls: 42.9% initial rhythm, 33.3% PEA, 9.5% VF/VT. TTM: initial rhythm: 26.1% vs. 36.4% for asystole, 34.8% vs. 31.8% for PEA, 13.0% vs. 9.1% for VF/VT	External cooling: four water-circulating gel-coated energy transfer pads placed on the patient’s back, abdomen, and both thighs. Surface area ranges: 0.60–0.77 m2 connected to an automatic thermostat	Median time in minutes to first ROSC (Mean, IQR) Control 24.0 (13.5, 42.0), Intervention external 25.0 (15.3, 39.3), internal 26.0 (12.3, 40.3) min	34 °C, after which, they were maintained at that temperature for 24 h before rewarming passively at 1 °C every 4 h (0.25 °C/h) to 36.5 °C	Sustained ROSC after CA for >30 min, Age: 18–80 years, hemodynamically stable, SBP > 90 mmHg	Comatose or unresponsive post-resuscitation, hypotension, pregnancy, premorbid status, bedbound and uncommunicative	Hospital discharge	Hospital discharge
Lopez-de-Sa et al. (2018) [23]	150	AED differed among groups: 32 °C group: 13.5% vs. 33 °C group: 34.7% vs. 34 °C group: 28.6%, (*p* = 0.03), Invasive coronary angiography: 61.5% vs. 81.6% vs. 73.5% (*p* = 0.08).	NR	32 °C 28.9 ± 15.9, 33 °C 26.3 ± 14.1, 34 °C 25.7 ± 11.6 min	32 °C, 33 °C, 34 °C	Signed consent, 18–80 years old, OHCA of cardiac cause, sustained ROSC, initial shockable cardiac rhythm, lack of meaningful response at arrival to hospital, non-traumatic CA, SBP > 90 Hgmm for 30 min post-ROSC	Traumatic/toxicological CA, pregnancy, Do Not Resuscitate order, femoral venous access contraindication, neuromuscular blocking agents prior to assesment, incomplete neurological evaluation, T° < 34 °C, current IVC filter, neurological illness, functional disabilities, intracranial bleeding, acute stroke, terminal illness, other RCT	3 months	3 months
Maynard et al. (2015) [30]	508	37% were discharged alive from the hospital, 58% had CPC = 1 or 2, 50% had MRS: slight disability or better	Rapid infusion of 2 L of 4 oC normal saline, Drug: Rapid infusion of cold normal saline	NR	33 °C vs. 36 °C	Successful resuscitation from OHCA (palpable pulse)	Traumatic cause	12 months	Hospital discharge (CPC)
Mérei et al. (2015) [35]	57	33% female, mean age: 62 years, CPR: 14.5 min, TH: 38%	2 L of intravascular saline at 4 °C vs. standard care and additional external cooling (cooler blocks over great blood vessels)	NR	33 °C vs. 36 °C for 24 h	Patients treated on the ICU of University of Pécs between June 2009–February 2012 after CPR, ROSC	NR	NR	1 month
Nordberg et al. (2019) [24]	677	Median time: 25 min after ROSC, median T°: intervention group: 34.6 °C vs. control group 35.8 °C (*p* < 0.001)	Mixture of air or oxygen and a liquid coolant via nasal catheters	Intervention group: 105 vs. 182 min in the control group (*p* < 0.001)	33 ± 1 °C	Witnessed CA, 18–80 years	Trauma cause, bleeding, drug overdose, cerebrovascular accident, drowning, smoke inhalation, electrocution, hanging, hypothermic, nasal anatomic barrier, Do Not Resuscitate order, terminal disease, pregnancy, coagulopathy, supplemental oxygen, EMS > 15 min	3 months	3 months (CPC 1 or 2)
Pang et al. (2016) [28]	21	Ventricular tachycardia/ventricular fibrillation: 33.3%, pulseless: 47.6%, asystole: 19.0%, CPR and ECLS duration: 25.7 min and 4.4 days	ECLS via percutaneous cannulation of the common femoral artery. Extracorporeal centrifugal pump, oxygenator and heat exchanger	NR	34 °C for 24 h	≥21 years, CA with ECLS, Ventricular fibrillation, Downtime <45 min, Comatose patients, Patients not responding to verbal after ROSC, ACLS < 60 min, mechanical ventilation	Responding to verbal commands after ROSC, CPR > 45 min, coagulopathy, pregnancy, premorbid status, uncommunicative, T° < 30 °C	6 months,	Hospital discharge (CPC) of 1–2
Scales et al. (2017) [20]	585	OHCA patients	Cold saline and ice packs applied to neck, axillae, and both groins and infusion of up to 2 L of cold saline at 4 °C	Median (IQR) cooling group 5.4 h (3.0–8.2) vs. control 4.8 h (2.8–7.7) (*p* = 0.45)	32–34 °C for 6 h	EMS-treated OHCA, age ≥18 years, ROSC of ≥5 min, SBP ≥ 100 mmHg, unresponsive to verbal stimuli, endotracheal intubation	Trauma, burn, or exposure HT, bleeding, sepsis, coagulopathy, Do Not Resuscitate order, pregnancy, or prisoner status	Hospital discharge	Hospital discharge

CA, cardiac arrest, CPR, cardiopulmonary resuscitation, TTM, therapeutic temperature management, CPC, Cerebral Performance Category, ECLS, extracorporeal life support, ROSC, return of spontaneous circulation, STEMI, ST-segment–elevation myocardial infarction, GCS, Glasgow Coma scale, RCT, randomized controlled trial, OHCA, Out-of-Hospital Cardiac Arrest, SBP, systolic blood pressure, PEA, pulseless electrical activity, VF/VT, Ventricular Fibrillation/tachycardia, AED, defibrillation with an automatic external defibrillator, IVC, Inferior Vena Cava, IQR, interquartile range, ECMO, patient is on extracorporeal membrane oxygenation, ICU, multidisciplinary intensive care unit, HT, hypothermia.

**Table 2 ijerph-18-11817-t002:** Characteristic of the selected RCTs in Pediatric patients.

Author (Year)	N	Baseline Characteristics	Cooling System	Mean Time Reaching Target Temperature	Target Temperature	Inclusion Criteria	Exclusion Criteria	Time of Evaluating Survival	Time of Evaluating Neurological Outcome
Fink et al. (2018) [29]	34	88% asphyxia, 82% OHCA, CPR duration: 20 (11.5, 30.0) min	Cooling blanket, cold saline infusion, cold packs, room T° regulation, and tepid bath	30 (27–33) (30.8 ± 5.0) h and 78 (75–79) (76.6 ± 24.1) h for the 24 and 72 h groups, respectively	33 ± 1°C	Arterial or venous catheter, GCS ≤ 8, HT initiated	Do Not Resuscitate status, pregnancy, contraindication for MRI, acute brain disease, brain death evaluation, metabolic disease affecting brain, active hemorrhage, coagulation defect	6 months	Neuron-specific enolase on day 7 (post-rewarming)
Meert et al. (2016) [21]	54	48 h-18 years, OHCA with CPR ≥ 2 min, required mechanical ventilation after ROSC	Surface cooling using a Blanketrol III cooling unit	Median (Q1, Q3) HT Group 5.8 (5.3, 6.3), Normothermia Group 6.1 (5.3, 7.0) (hours)	33 °C for 48 h	<1 year at OHCA and ALTE as the aetiology of arrest	Inability to be randomised within six hours of return of circulation, GCS = 5 or 6.21, aggressive treatment, trauma cause CA.	28 days	12 months (VABS-II)
Moler et al. (2015) [31]	295	Ventricular fibrillation or ventricular tachycardia: 8%	Pharmacological HT (paralyzed and sedated), Blanketrol III temperature management unit (blankets applied anteriorly and posteriorly)	HT group: 5.9 h (IQR: 5.2–6.7) vs. normothermia group: 5.8 h (IQR: 5.0–6.4)	33.0 °C (32.0–34.0) for 48 h	48 h, 18 years of age, CA with CPR = 2 min and mechanical ventilation after ROSC	Inability to randomization within 6 h after ROSC, GCS motor-response subscale: 5 or 6	12 months	12 months (VABS-II)
Moler et al. (2017) [32]	329	Median age: 1 year, 60% male, 91% preexisting medical condition, 57% bradycardia, 10% fibrillation or ventricular tachycardia, 93% hospital CA	Blanketrol III temperature-management unit applied anteriorly and posteriorly	CPR median duration 22.0 min (IQR 7.0–47.0)	33.0 °C (32.0–34.0) for 48 h	48 h, 18 years of age CA in a hospital, RCP > 2 min with mechanical ventilation after ROSC	GCS motor-response subscale = 5 or 6, inability to randomization 6 h after ROSC, bleeding, a preexisting illness, life expectancy <12 months, aggressive treatment	12 months	12 months (VABS-II)
Silverstein et al. (2016) [33]	295	CA etiology: respiratory (66%), initial EKG rhythm: asystole (66%), bradycardia (34%), pulseless electrical activity (36%)	Blanketrol III temperature-management unit applied anteriorly and posteriorly	NR	33 °C ± 1 °C for 48 h	CPR >2 min with ROSC, 48 h, 18 years, continuous mechanical ventilation requirement unplanned CA	Speaking barriers, ECMO when CA ≥2 ug/kg/minute infusion of epinephrine or norepinephrine, GCS = 5–6, prior CA with CPR > 2 min, life expectancy < 12 months, Do Not Resuscitate order	1 year	12 months (VABS-II score ≥70)

CA, cardiac arrest, VABS-II, Vineland Adaptive Behavior Scales, Second Edition, GCS, Glasgow Coma scale, OHCA, Out-of-Hospital Cardiac Arrest, IQR, interquartile range, ECMO, patient is on extracorporeal membrane oxygenation, CRP, cardiopulmonary resucitation, ROSC, return of spontaneous circulation, NR, non reported, HT, hypothermia.

**Table 3 ijerph-18-11817-t003:** Main results of included RCTs in adult population.

Author (Year)	N	Aim	Intervention	Results
Bernard et al. (2016) [19]	1198	To measure survival rate at hospital discharge and time to return to a spontaneous circulation after CA	CPR + TTM vs. CPR + 2 L of cold saline perfusion during an out-of-hospital CRP	TTM during CPR has no effect on survival rate and may decrease spontaneous circulation recovery in patients with shockable rhythms
Cronberg et al. (2015) [25]	939	To evaluate the effects of TTM on survival rate, cognitive status, functionality, and quality of life	TTM at 33–36 °C. CPC, mRS, MMSE, ALFI, IQCODE, and SF-36 scales were measured at 6 months of follow-up in CA survivors	Survival rate and quality of life were similar in both groups
Kirkegaard et al. (2017) [26]	355	To evaluate effectiveness of 48 and 24 h of TTM	48 h of TTM at 33 °C vs. 24 h of TTM at 33 °C + 0.5 °C per hour rewarming	There were no differences in mortality between groups. Patients in 48 h of TTM group showed more adverse events
Lascarrou et al. (2019) [22]	584	To measure the effectiveness of TTM in patients in a non-shockable rhythm	Patients were assigned to 33 °C TTM for 24 h vs. normothermia treatment at 37 °C	TTM group patients showed higher survival and better neurological status rates than the normothermia group
Lilja et al. (2015) [27]	652	To assess 6-month neurological status in patients who received TTM at 33–36 °C [19] compared to MINOCA patients.	RBMT-3, FAB, Symbols and Digits Test, and HADS scales were used to compare neurological status between groups.	Cognitive function was comparable in CA survivors and MINOCA groups
Look et al. (2018) [34]	87	To analyze survival rate after hospital discharge and neurological status in survivors of CA after undergoing intravascular and surface cooling devices for TTM.	Patients were randomized to intravascular and surface cooling devices for TTM or to non- therapeutic hypothermia treatment.	Intravascular TTM showed more accurate temperature control, and non-significant higher rates of survival were found in comparison with surface cooling.
Lopez-de-Sa et al. (2018) [23]	150	To compare the effect of different temperatures of TTM on neurological status in out-of-hospital CA	Comatose patients underwent to TTM of 32 °C, 33 °C, and 34 °C for 24 h.	There was no significant impact on neurological status and survival rate.
Maynard et al. (2015) [30]	508	Effect of prehospital induction of mild hypothermia on 3-month neurological status and 1-year survival among out-of-hospital CA treated with out-of-hospital TTM.	Cerebral Performance Category (CPC) and Modified Rankin Scale (mRS) were measured by telephone call to compare neurological status between patients treated with out-of-hospital TTM (2 L of intravascular saline at 4 °C after CPR) vs. standard care after resuscitation from CA.	TTM showed no improvement on the neurological status and survival rate.
Mérei et al. (2015) [35]	57	To evaluate 30-day survival, TTM efficacy and serum levels of S100B protein as a prognostic biomarker.	Blood samples of 20 CA patients randomly selected from TTM of 32–34 °C and standard care groups were taken at admission and after 12 and 36 h after recovery to measure S100B levels.	TTM showed no improvement on survival and S100B serum levels.
Nordberg et al. (2019) [24]	677	To determine whether prehospital trans-nasal evaporative intra-arrest cooling improves survival with good neurologic outcomes compared with cooling initiated after hospital arrival.	Patients were randomly assigned to receive trans-nasal evaporative intra-arrest cooling or standard care. Both groups received TTM at 32 °C to 34 °C for 24 h.	Trans-nasal evaporative intra-arrest cooling did not result in a statistically significant improvement in survival with good neurologic outcome, however, those patients reached target T° earlier.
Pang et al. (2016) [28]	21	To evaluate safety and efficacy of TTM in patients undergoing ECLS after CA.	Comparison of a control group of patients with ECLS treated with normothermia (37 °C) vs. TTM at 34 °C for 24 h.	TTM can be applied safely in ECLS patients. No significant differences were found in terms of survival or neurological status.
Scales et al. (2017) [20]	585	To evaluate if pre-hospital cooling (target temperature of 32–34 °C within 6 h of hospital arrival) leads to higher rates of successful TTM.	Patients were randomized to receive prehospital cooling (initiated 5 min after ROSC) or usual resuscitation and transport. Both groups received TTM in the critical care unit.	Prehospital cooling after ROSC did not increase rates of achieving a target temperature of 32–34 °C within 6 h of hospital arrival, but it was safe and increased application of TTM in hospital.

CA, cardiac arrest, CPR, cardiopulmonary resuscitation, TTM, therapeutic temperature management, CPC, Cerebral Performance Category, mRS, Modified Ranking Scale, MMSE, Mini-Mental State Examination, ALFI, Adult Lifestyles and Function Interview, IQCODE, Informant Questionnaire of Cognitive Decline in the Elderly, SF-36, Short Form-36 Health Survey, MINOCA, myocardial infarction with non-obstructive coronary arteries, RBMT-3, Rivermead Behavioral Memory Test, FAB, frontal assessment battery, HADS, The Hospital Anxiety and Depression Scale, ECLS, extracorporeal life support, ROSC, return of spontaneous circulation.

**Table 4 ijerph-18-11817-t004:** Main results of the included RCTs in Pediatric patients.

Author (Year)	N	Aim	Intervention	Results
Fink et al. (2018) [29]	34	To measure effect of TTM at 33 °C in pediatric patients for 24 h vs. 72 h on 6-month mortality and different biomarkers.	Children with CA were randomized to TH for 24 or 72 h. Serum was collected twice daily on days 1–4 and once on day 7. Mortality was assessed at 6 months.	Serum NSE and S100b protein concentration was increased in the 24 h group. Significant difference was not shown in mortality.
Meert et al. (2016) [21]	54	To explore the safety and efficacy of TTM amongst infants with out-of-hospital CA.	Patients were allocated to TTM at 33 °C or at 36.8 °C (normothermia) for 48 h within six hours of return of circulation. 12-month survival was measured though VABS-II scale.	No differences in survival rates or neurologic status were found between groups.
Moler et al. (2015) [31]	295	To assess TH efficacy on survival and functional outcome in out-of-hospital CA pediatric patients.	TH at 33.0 °C was compared to therapeutic normothermia at 36.8 °C. One-year survival with a good neurobehavioral was estimated thought the VABS-II scale.	Survival at 12 months did not differ significantly between groups.
Moler et al. (2017) [32]	329	To assess TH efficacy on survival and functional outcome in out-of-hospital CA pediatric patients.	TH at 33.0 °C was compared to therapeutic normothermia at 36.8 °C. One-year survival with a functional outcome was estimated through the VABS-II scale.	Trial was stopped because of assessment of futility before attainment of the target trial enrollment.
Silverstein et al. (2016) [33]	295	To assess functionality in out-of-hospital CA pediatric patients.	TH at 33 °C was compared to normothermia at 36.8 °C. Functionality was measured though VABS-II, PCPC, and POPC scales.	No differences were found between the groups in terms of functional status.

TTM, therapeutic temperature management, CA, cardiac arrest, TH, therapeutic hypothermia, NSE, neuron-specific enolase, VABS-II, Vineland Adaptive Behavior Scales, Second Edition, PCPC, Pediatric Cerebral Performance Category, POPC, Pediatric Overall Performance Category.

## Data Availability

Not applicable.

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
