# Peer review of "The Effect of Therapeutic Hypothermia after Cardiac Arrest on the Neurological Outcome and Survival—A Systematic Review of RCTs Published between 2016 and 2020"

_ijerph, 2021, doi:10.3390/ijerph182211817_

Round 1
Reviewer 1 Report
Ιt would be acceptable for publication
Author Response
Reviewers' comments:
Reviewer: 1
Manuscript ID: ijerph-1440221
Type of manuscript: Review
Title: The effect of therapeutic hypothermia after cardiac arrest on the neurological outcome and survival. The authors would like to express their sincere thanks to the reviewer whose suggestions will improve our manuscript.
Minor comments:
Remark 1: Ιt would be acceptable for publication Thank you very much for taking the time to review this work. It is a pleasure to receive your kind comments with the aim of improving the manuscript.
Reviewer 2 Report
Ad Selection flow diagram, the numbers listed are not clear to me. Records initially identified is mentioned as n=714, however, calculating the numbers mentioned to the right n=606.
Additionally the number of records excluded (Not RCTs) is not mentioned. Further on "Studies potentially relevant", which is described as n=211 does not match with the number of studies included after "Studies excluded after title and abstract review" have been subtracted. (211-189= 22 not 17)
Please clarify!
Author Response
Reviewer: 2
Manuscript ID: ijerph-1440221
Type of manuscript: Review
Title: The effect of therapeutic hypothermia after cardiac arrest on the neurological outcome and survival.
The authors would like to express their sincere thanks to the reviewer whose suggestions will improve our manuscript. We appreciate the comments that have been made to us. They are very pertinent and valuable comments. We believe that after addressing them, the results obtained may be more robust. Thank you for your time.
Remark 1: Ad Selection flow diagram, the numbers listed are not clear to me. Records initially identified is mentioned as n=714, however, calculating the numbers mentioned to the right n=606. Additionally the number of records excluded (Not RCTs) is not mentioned. Further on "Studies potentially relevant", which is described as n=211 does not match with the number of studies included after "Studies excluded after title and abstract review" have been subtracted. (211-189= 22 not 17)
Please clarify!
We agree with your comment. The previous version gave inaccurate information. The section has been changed accordingly. This was an error in the study identification stage, as the articles found in the CUIDEN database had been included without entering the search restrictions (in the end, n= 0 articles were found in this database).

This manuscript is a resubmission of an earlier submission. The following is a list of the peer review reports and author responses from that submission.
Round 1
Reviewer 1 Report
Thank you very much for submitting your work
howeever, it is not well worked out and very confusing. I think it yould be helpfull to introduce a table with name of original, number of participants, baselinecharacteristics, used cooling system (intracorporal vs extra, active vs passive), mean time reaching target temperature, target temperature (you can not compare a TTM of 32 with 36°C and a TTM for 24 hours with TTM for 72 h ...... )
Inclusion and exclusion of each
time of evaluating survival
time of evaluating neurol outcome (Mord Rank vs CPC score)
Definitions of good and poor outcome
Tirel should be changed in survival and neurological outcome, while neurological status has another meaning.
However overall it is very confusing without any new impact
Reviewer 2 Report
Attached is the file.

Reviewer 3 Report
The authors provide a systematic review about the effect of therapeutic hypothermia on neurological status and survival in patients who have suffered cardiorespiratory arrest and remain conscious after the recovery of spontaneous circulation.
This issue has been discussed, in the form of meta-analysis/systematic review, by many recent articles (e.g. doi: 10.1016/j.resuscitation.2021.01.029, doi: 10.1016/j.amjcard.2020.07.038).
Apart from that, the article has many methodological problems:
In the introduction section:
I believe that it should be smaller.
In the methods section:
The authors should report why they searched only the last 5 years and why they used the PEDro scale and not ROB2.
In the results section:
It is very small, as they report many parts of results in the methods or in the discussion sections (e.g. “ After the first screening, 714 studies were collected. Once duplicates were removed (N=260), 454 studies were selected as potentially relevant and reviewed by title and abstract. 17 studies were finally included according to the inclusion criteria for full reading and data extraction (Figure 1).”, “Survival rate None of the studies included in this review showed significant differences in survival rate.”)